# Contrastive Learning of Sentence Embeddings from Scratch

**Junlei Zhang**[1,2*]    **Zhenzhong Lan**[2]    **Junxian He**[†3]
[1]Zhejiang University    [2]School of Engineering, Westlake University
[3]The Hong Kong University of Science and Technology
{zhangjunlei,lanzhenzhong}@westlake.edu.cn, junxianh@cse.ust.hk

## Abstract

Contrastive learning has been the dominant approach to train state-of-the-art sentence embeddings. Previous studies have typically learned sentence embeddings either through the use of human-annotated natural language inference (NLI) data or via large-scale unlabeled sentences in an unsupervised manner. However, even in the case of 1;unlabeled data, their acquisition presents challenges in certain domains due to various reasons. To address these issues, we present SynCSE, a contrastive learning framework that trains sentence embeddings with synthesized data. Specifically, we explore utilizing large language models to synthesize the required data samples for contrastive learning, including (1) producing positive and negative annotations given unlabeled sentences (*SynCSE-partial*), and (2) generating sentences along with their corresponding annotations from scratch (*SynCSE-scratch*). Experimental results on sentence similarity and reranking tasks indicate that both SynCSE-partial and SynCSE-scratch greatly outperform unsupervised baselines, and SynCSE-partial even achieves comparable performance to the supervised models in most settings.[1]

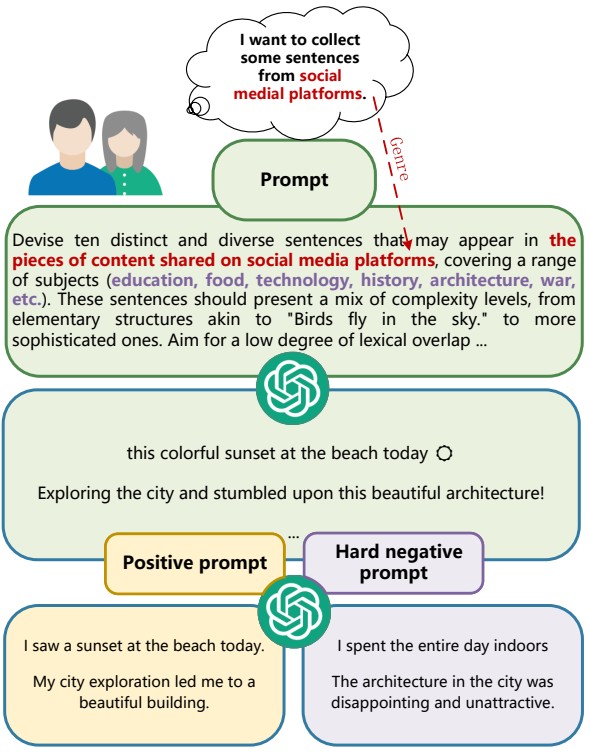

Figure 1: An overview of the data synthesis process of SynCSE-scratch. We specify a desired domain and genre, and our framework will generate diverse unlabeled data for that domain along with their positive and negative annotations.

## 1 Introduction

The objective of sentence representation learning is to derive sentence embeddings that can benefit a wide range of downstream tasks, including reranking (Lee et al., 2021; Barker et al., 2021), natural language understanding (Cer et al., 2018), and retrieval (Misra et al., 2016; Thakur et al., 2021; Wang et al., 2022a). Methods built on contrastive learning, such as SimCSE (Gao et al., 2021) and PromCSE (Jiang et al., 2022b), have dominated the field due to their competitive performance (Zeng et al., 2022; Limkonchotiwat et al., 2022; Wu et al., 2022a; Wang et al., 2022c; He et al., 2023).

Contrastive learning trains sentence representations through distinguishing positive samples from negative ones. In this framework, the quality of these positive and negative annotations plays a critical role. Supervised approaches typically gather these annotations from labeled natural language inference (NLI) datasets (Jiang et al., 2022a; Limkonchotiwat et al., 2022) – however, such sources are generally unavailable for most settings, and manually creating them is cost-prohibitive. As a result, unsupervised methods that solely rely on unlabeled sentences attract significantly more attention re-

---

*Work done during Junlei's visit to HKUST.
†Corresponding author.

[1]Code and the synthesized datasets are available at https://github.com/hkust-nlp/SynCSE.

cently (Gao et al., 2021; Zhou et al., 2022; Wu et al., 2022a) – they mostly develop methods to automatically obtain positive and negative samples to facilitate contrastive learning. A representative example is SimCSE (Gao et al., 2021), which leverages perturbed hidden states as the positive samples and in-batch sentences as negatives to perform contrastive learning. To differentiate between in-batch negatives and the annotated negatives, the latter are often termed "hard negatives", which have proven to be significantly advantageous in enhancing sentence embeddings (Wang et al., 2022b,c).

Despite considerable advances in recent years, the performance of these unsupervised methods still falls short when compared to their supervised counterparts. Moreover, the unavailability of large-scale unlabeled data for the targeted domain often poses additional limitations to these approaches. To overcome these challenges, we introduce SynCSE, an unsupervised contrastive framework that trains sentence embeddings with synthesized data. Concretely, we propose to prompt large language models (LLMs) such as ChatGPT (OpenAI, 2022) to synthesize the samples needed for contrastive learning. This is inspired by recent successes of prompting large language models (LLMs) to perform various tasks (Chung et al., 2022; Ouyang et al., 2022; OpenAI, 2023), especially the superior performance of LLMs over crowd-workers on text annotation (Gilardi et al., 2023). We investigate two variants of SynCSE in this work that correspond to two practical scenarios: (1) *SynCSE-partial*, where large-scale unlabeled sentences are available and LLMs are prompted to produce positive and hard negative annotations, and (2) *SynCSE-scratch*, where large-scale unlabeled sentences are not available, prompting LLMs to generate sentences and their corresponding annotations from scratch. The latter represents a particularly challenging yet practical scenario where we aim to learn sentence embeddings without any data samples.

We conduct comprehensive experiments on the standard Semantic Textual Similarity (STS) benchmark, along with four reranking tasks and four domain adaptation tasks. Our results demonstrate that both SynCSE-partial and SynCSE-scratch substantially outperform the unsupervised baselines in all cases – for example, SynCSE-partial and SynCSE-scratch exceed the unsupervised SimCSE baseline by 5.37 and 4.18 absolute points respectively on STS. Particularly, SynCSE-partial often equals its supervised counterpart on STS, marking the first instance of an unsupervised method matching supervised results on this benchmark. We release our synthesized datasets to facilitate further research to learn better sentence embeddings.

## 2 SynCSE

### 2.1 Background

We base our approach on the formulation of SimCSE (Gao et al., 2021), which is one of the most common and effective contrastive learning frameworks to learn sentence embeddings. Formally, we denote the unlabeled sentence as $x_i$ and its positive sample as $x_i^+$. Let $\boldsymbol{h}_i$ and $\boldsymbol{h}_i^+$ denote the representations of $x_i$ and $x_i^+$ respectively, then the unsupervised SimCSE loss is defined as:

$$-\log \frac{e^{sim(\boldsymbol{h}_i, \boldsymbol{h}_i^+)/\tau}}{\sum_{j=1}^{M} e^{sim(\boldsymbol{h}_i, \boldsymbol{h}_j^+)/\tau}}, \qquad (1)$$

where M denotes the mini-batch's size, $\tau$ is a temperature hyperparameter, and $sim(\cdot, \cdot)$ stands for a similarity function. Unsupervised SimCSE passes the same $x_i$ twice to the encoder to form $(\boldsymbol{h}_i, \boldsymbol{h}_i^+)$ pairs due to random dropout, and other sentences within the same mini-batch are considered as negative samples as shown in Eq. 1. Supervised SimCSE further extends $(x_i, x_i^+)$ with hard negative samples $x_i^-$ to constitute the triplet datasets $\{x_i, x_i^+, x_i^-\}_{i=1}^{N}$ and define the supervised loss:

$$-\log \frac{e^{sim(\boldsymbol{h}_i, \boldsymbol{h}_i^+)/\tau}}{\sum_{j=1}^{M} (e^{sim(\boldsymbol{h}_i, \boldsymbol{h}_j^+)/\tau} + e^{sim(\boldsymbol{h}_i, \boldsymbol{h}_j^-)/\tau})}. \qquad (2)$$

In supervised SimCSE, the $(x_i, x_i^+, x_i^-)$ triplets are typically from annotated NLI datasets, where $x_i$ is the premise, $x_i^+$ and $x_i^-$ are the entailment and contradiction hypotheses. Supervised SimCSE significantly outperforms the unsupervised one due to the enhanced quality of positive and hard negative samples. However, such annotated data are typically unavailable in most settings, and manually annotating triplets $(x_i, x_i^+, x_i^-)$ can be resource-intensive, rendering unsupervised approaches the most promising choices in practice. In this work, we focus on the supervised loss in Eq. 2, but synthesize $(x_i^+, x_i^-)$ given $x_i$ or even generate $(x_i, x_i^+, x_i^-)$ triplets from scratch, aiming to approach the performance of supervised models with an unsupervised method. We describe our data synthesis process next.

| Hard negative prompts pools |
|---|
| **Prompt1**: Revise the provided sentence by swapping, changing, or contradicting some details in order to express a different meaning, while maintaining the general context and structure. |
| **Prompt2**: Generate a slightly modified version of the provided sentence to express an opposing or alternate meaning by changing one or two specific elements, while maintaining the overall context and sentence structure. |
| **Prompt3**: Transform the input sentence by adjusting, altering, or contradicting its original meaning to create a logical and sensible output sentence with a different meaning from the input sentence. |
| **Prompt4**: Generate a sentence that conveys a altering, contrasting or opposite idea to the given input sentence, while ensuring the new sentence is logical, realistic, and grounded in common sense. |

Table 1: Hard negative prompts pools. During the generation of hard negative samples, a hard negative prompt is randomly sampled each time.

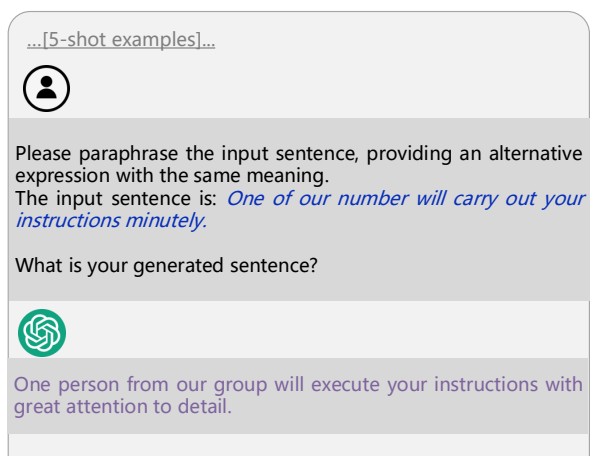

Figure 2: Few-shot examples of generating positive examples of the input sentence. We adopt 5-shot for generation.

## 2.2 Data Synthesis from ChatGPT

We propose to prompt ChatGPT (OpenAI, 2022) to synthesize the required data in contrastive learning, inspired by recent successes of prompting LLMs to fulfill multiple tasks (Chung et al., 2022; OpenAI, 2023). Concretely, we introduce two variants of SynCSE: (1) SynCSE-partial which synthesizes $(x_i^+, x_i^-)$ given $x_i$, and (2) SynCSE-scratch which synthesizes $(x_i, x_i^+, x_i^-)$ from scratch. SynCSE-scratch is practically useful since large-scale unlabeled data are not always available in the domain of interest due to copyright restrictions, data distribution issues, or messy formats. We describe these two variants below. In general, using SynCSE-scratch as an example, the complete data generation process includes two parts: (1) generating unlabeled sentences in the target domain; (2) generating

positive/hard negative labels with prompt/example pools.

## 2.3 SynCSE-partial

**Synthesizing positive and hard negative examples:** We prompt ChatGPT in a few-shot setting to annotate positive and hard negative samples given a sentence $x_i$, an illustrative example is shown in Figure 2. The structure of the prompts for generating positive and hard negative examples remains the same; the only difference lies in the prompts. In our implementation with the ChatGPT model, we have designed a few-shot prompt in a multi-turn chat format.

**Example and prompt pools:** A significant challenge in creating synthetic datasets lies in enhancing the dataset's diversity. Ye et al. (2022b) suggested that merely increasing the size of the synthetic dataset might not lead to better performance, with one reason being the lack of diversity. Datasets labeled by groups of annotators can naturally help to mitigate this problem due to the variance in understanding and interpretation of prompts among different annotators. This variance results in diverse outputs, even for the same input. For example, Williams et al. (2018) utilized 387 annotators to create the MultiNLI dataset. Even with the same prompt, these annotators provided varied outputs due to their individual understanding of the prompt and their unique world knowledge, leading to a more diverse dataset. In an attempt to mimic this variation among different annotators, we employ example pools and prompt pools. Specifically, we designed four types of positive/hard negative prompts (an example of hard negative prompts are

showed in Table 1) and 18 few-shot exemplars for each of the prompt (generated using GPT-4). During each data generation process, we sample one prompt and five exemplars to construct a distinct input prompt. Details of these pools can be found in Appendix A.

## 2.4 SynCSE-scratch

Creating a synthetic dataset from scratch, where the necessary unlabeled sentences for annotation are absent, presents a substantial challenge. We address this problem in two stages: initially, we generate unlabeled sentences, and subsequently, we apply the procedure discussed in §2.3 to annotate positive and hard negative samples of these sentences.

To ensure data diversity during the generation of unlabeled sentences, we employ a strategy that specifies the genres and topics when generation, combined with the utilization of example and prompt pools. This strategy is intended to minimize repetition and redundancy between the new data and the generated data so far. More specifically, as illustrated in Figure 1, given a text genre, we randomly select six topics from a pre-defined list to be included in the prompt (the list of genres and topics used in this paper can be found in Appendix B). The term "etc." in the prompt ensures that the generated sentences are not strictly limited to these six topics. We adopt one-shot prompting to generate several sentences at once. As long as given different genres or topics when adding data compared to the existing data, the added data will likely have low redundancy with the existing data, thereby enhancing the overall diversity of the dataset. The examples used for generating raw sentences were produced by GPT-4.

## 3 Experiment

## 3.1 Training

We evaluate three different settings in the experiments, including SynCSE-partial, SynCSE-scratch, as well as a combination of SynCSE-scratch with existing annotated datasets in a supervised setting. While both SynCSE-partial and SynCSE-scratch represent unsupervised settings, in the combination setting we augment previous annotated datasets with the synthesized data produced in SynCSE-scratch, to examine whether SynCSE-scratch could provide help for a supervised scenario as well.

We refer to the NLI dataset (MNLI+SNLI) used

by SimCSE as SimCSE_NLI. In the creation of the SynCSE-partial dataset, for a fair comparison, we utilized the unlabeled sentences $x$ from Sim-CSE_NLI, and generated positive/hard negative examples for them using the algorithm detailed in §2.3. For SynCSE-scratch, we generate the same number of examples as in the SynCSE-partial case, as detailed in §2.4. While our method can easily scale up the dataset, for a fair comparison, we ensure the data volume used for SynCSE-scratch and SynCSE-partial is equivalent to that of Sim-CSE_NLI. For the combination of the SynCSE-scratch and SimCSE_NLI datasets, we simply merge these two datasets to evaluate whether our generated dataset can aid the manually annotated one.

Given that SimCSE serves as a general method in contrastive learning, we consistently use SimCSE as the backbone method for SynCSE. We note that SynCSE is general and could be combined with more advanced algorithms as well, such as with PromCSE (Jiang et al., 2022b) and CARDS (Wang et al., 2022c). We emphasize that, after training the models on the NLI dataset, we freeze the models and directly evaluate our embeddings on all the different tasks and setting below – we do not specifically train sentence embeddings on each setting separately. For the STS and transfer learning tasks, we use the same hyperparameters as Sim-CSE. Since SimCSE did not conduct reranking experiments, we directly use the default parameters of MTEB (Muennighoff et al., 2023) to evaluate embeddings on the reranking tasks.

## 3.2 Evaluation Settings

**Semantic Textual Similarity Tasks:** Following the procedure outlined in SimCSE, we evaluate our model, trained on the synthetic NLI dataset, across seven semantic textual similarity (STS) tasks: STS 2012-2016 (Agirre et al., 2012, 2013, 2014, 2015, 2016), the STS Benchmark (Cer et al., 2017), and SICK Relatedness (Marelli et al., 2014). It is important to note that no data from these STS tasks were used during training. Our model was trained solely on our synthetic NLI dataset. The sentence embeddings, which we evaluate on the STS tasks, are obtained from the $[CLS]$ representation. During the training process, we average the development scores from the STS Benchmark and SICK Relatedness to form the evaluation matrix. This matrix is used to select the best models. The other hyper-

| Model | Method | STS12 | STS13 | STS14 | STS15 | STS16 | STSb | SICK-R | Avg |
|---|---|---|---|---|---|---|---|---|---|
| | *Unsupervised methods* | | | | | | | | |
| RoBERTa-base | unsup-SimCSE[†] | 70.16 | 81.77 | 73.24 | 81.36 | 80.65 | 80.22 | 68.56 | 76.57 |
| | RankCSE$_{listNet}^{§}$ | 72.88 | 84.50 | 76.46 | 84.67 | **83.00** | 83.24 | 71.67 | 79.49 |
| | RankCSE$_{listMLE}^{§}$ | 72.74 | 84.24 | 75.99 | 84.68 | 82.88 | 83.16 | 71.77 | 79.35 |
| | L2P-CSR[♠] | 74.97 | 83.63 | 78.28 | 84.86 | 82.03 | 82.77 | 71.26 | 79.69 |
| | PromptRoBERTa[††] | 73.94 | **84.74** | 77.28 | 84.99 | 81.74 | 81.88 | 69.50 | 79.15 |
| | PCL[‡‡] | 71.54 | 82.70 | 75.38 | 83.31 | 81.64 | 81.61 | 69.19 | 77.91 |
| | CARDS[°] | 72.49 | 84.09 | 76.19 | 82.98 | 82.11 | 82.25 | 70.65 | 78.68 |
| | ConPVP[•] | 73.20 | 83.22 | 76.24 | 83.37 | 81.49 | 82.18 | 74.59 | 79.18 |
| | SynCSE-partial (SimCSE based) | **76.11** | 84.49 | **79.61** | **85.26** | 82.60 | **83.94** | **81.57** | **81.94** |
| | SynCSE-scratch (SimCSE based) | 74.61 | 83.76 | 77.89 | 85.09 | 82.28 | 82.71 | 78.88 | 80.75 |
| RoBERTa-large | unsup-SimCSE[†] | 72.86 | 83.99 | 75.62 | 84.77 | 81.80 | 81.98 | 71.26 | 78.90 |
| | RankCSE$_{listNet}^{§}$ | 73.23 | 85.08 | 77.50 | 85.67 | 82.99 | 84.20 | 72.98 | 80.24 |
| | RankCSE$_{listMLE}^{§}$ | 73.40 | **85.34** | 77.25 | 85.45 | 82.64 | 84.14 | 72.92 | 80.16 |
| | L2P-CSR[♠] | 73.65 | 84.08 | 78.29 | 85.36 | 82.15 | 83.70 | 73.47 | 80.10 |
| | PCL[‡‡] | 73.76 | 84.59 | 76.81 | 85.37 | 81.66 | 82.89 | 80.33 | 79.34 |
| | CARDS[°] | 74.63 | 86.27 | 79.25 | 85.93 | 83.17 | 83.86 | 72.77 | 80.84 |
| | ConPVP[•] | 74.75 | 84.09 | 77.88 | 83.13 | 83.44 | 83.64 | 74.31 | 80.18 |
| | SynCSE-partial (SimCSE based) | **76.03** | 84.27 | 80.03 | 85.37 | 83.62 | 84.26 | **81.14** | 82.10 |
| | SynCSE-scratch (SimCSE based) | 75.45 | 85.01 | **80.28** | **86.55** | **83.95** | **84.49** | 80.61 | **82.33** |
| | *Supervised methods* | | | | | | | | |
| RoBERTa-base | sup-SimCSE[†] | 76.53 | 85.21 | 80.95 | 86.03 | 82.57 | 85.83 | 80.50 | 82.52 |
| | sup-SimCSE | 75.61 | 85.19 | 79.58 | 85.85 | 82.39 | 85.30 | 80.39 | 82.04 |
| | PromptRoBERTa[††] | 76.75 | **85.93** | **82.28** | 86.69 | 82.80 | **86.14** | 80.04 | **82.95** |
| | PrompCSE + EH[♡] | 75.96 | 84.99 | 80.44 | **86.83** | 81.30 | 84.40 | 80.96 | 82.13 |
| | SynCSE-scratch + SimCSE_NLI | **76.79** | 84.93 | 80.14 | 86.28 | 83.38 | 85.75 | **81.02** | 82.61 |
| RoBERTa-large | sup-SimCSE[†] | 77.46 | 87.27 | 82.36 | 86.66 | 83.93 | 86.70 | 81.95 | 83.76 |
| | sup-SimCSE | 76.62 | 86.90 | 82.05 | 86.10 | 83.97 | 86.10 | 82.04 | 83.40 |
| | PromCSE + EH[♡] | **79.56** | **88.97** | **83.81** | **88.08** | 84.96 | **87.87** | 82.43 | **85.10** |
| | SynCSE-scratch + SimCSE_NLI | 77.13 | 87.61 | 82.82 | 87.67 | **85.66** | 87.22 | **82.45** | 84.37 |

Table 2: Results on the STS benchmark. Spearman's correlation is reported. The "unsup-" and "sup-" correspond to unsupervised and supervised settings, respectively. "†": results from (Gao et al., 2021); "§": results from (Liu et al.); "♠": results from (Zhou et al., 2023); "††": results from (Jiang et al., 2022a); "‡‡": results from (Wu et al., 2022a); "°": results from (Wang et al., 2022c); "•": results from (Zeng et al., 2022); "♡": results from (Jiang et al., 2022b). The term "SynCSE-scratch + SimCSE_NLI" represents our synthetic data combined with the NLI dataset used in SimCSE. The SynCSE-partial/scratch experiments were implemented on the basis of SimCSE. Some baselines did not conduct some experimental setups. We report the results that exist in their papers.

parameters are kept consistent with those used in SimCSE.

**Reranking tasks:** We further evaluate the synthetic dataset on four reranking tasks: AskUbuntuDupQuestions (Lei et al., 2016), MindSmallReranking (Wu et al., 2020), SciDocsRR (Cohan et al., 2020), and StackOverflowDupQuestions (Liu et al., 2018). We directly evaluate the model, which is frozen after training on the NLI dataset, on reranking tasks, without using the training sets of reranking tasks. The resulting ranking is scored for each query and averaged across all queries. In line with the methodology of MTEB (Muennighoff et al., 2023), we utilize Mean Average Precision (MAP) as the primary metric.

**Baselines:** We compare our approach with state-of-the-art sentence embedding learning methods: RankCSE (Liu et al.), L2P-CSR (Zhou et al., 2023), PCL (Wu et al., 2022a), CARDS (Wang et al., 2022c), ConPVP (Zeng et al., 2022), and PromptRoBERTa (Jiang et al., 2022a). While we base our approach on SimCSE, we emphasize that our approach *is orthogonal to the baseline algorithms* and our synthesized datasets may be combined with them to further boost the performance. We directly report the results from their respective papers.

### 3.3 Semantic Texual Similarity

**Main results:** As shown in Table 2, Both SynCSE-partial and SynCSE-scratch significantly

| Model | Method | AskU. | MindSmall | SciDocsRR | StackO. | Avg |
|---|---|---|---|---|---|---|
| *Unsupervised methods* | | | | | | |
| RoBERTa-base | unsup-SimCSE | 52.78 | 29.91 | 65.96 | 39.25 | 46.95 |
| | CARDS | 52.94 | 27.92 | 64.62 | **41.51** | 46.75 |
| | PCL | 51.85 | 27.92 | 64.70 | 41.18 | 46.41 |
| | SynCSE-partial (SimCSE based) | **53.95** | 29.97 | 65.21 | 37.84 | 46.74 |
| | SynCSE-scratch (SimCSE based) | 53.27 | **30.29** | **67.55** | 39.39 | **47.63** |
| RoBERTa-large | unsup-SimCSE | 55.10 | 29.23 | 68.54 | 42.56 | 48.86 |
| | CARDS | 53.83 | 29.07 | 68.26 | **43.24** | 48.60 |
| | PCL | 53.43 | 28.56 | 66.06 | 41.54 | 47.40 |
| | SynCSE-partial (SimCSE based) | 54.78 | 30.23 | 68.90 | 38.28 | 48.05 |
| | SynCSE-scratch (SimCSE based) | **55.48** | **30.27** | **70.85** | 40.00 | **49.15** |
| *Supervised methods* | | | | | | |
| RoBERTa-base | sup-SimCSE | 52.55 | 29.87 | **68.43** | 37.52 | 47.09 |
| | SynCSE-scratch + SimCSE_NLI | **52.74** | **30.40** | 67.65 | **38.17** | **47.24** |
| RoBERTa-large | sup-SimCSE | 54.72 | **30.89** | **71.69** | 38.24 | 48.89 |
| | SynCSE-scratch + SimCSE_NLI | **55.26** | 30.40 | 71.53 | **39.84** | **49.26** |

Table 3: Results on the reranking benchmark. Mean Average Precision (MAP) is reported.

| Dataset | STS12 | STS13 | STS14 | STS15 | STS16 | STSb | SICK-R | Avg |
|---|---|---|---|---|---|---|---|---|
| GenSE [‡] | 72.09 | **85.24** | **79.84** | 83.25 | **82.88** | 83.24 | 75.33 | 80.27 |
| DINO [†] | 70.27 | 81.26 | 71.25 | 80.49 | 77.18 | 77.82 | 68.09 | 75.20 |
| SynCSE-partial (SimCSE based) | **76.11** | 84.49 | 79.61 | **85.26** | 82.60 | **83.94** | **81.57** | **81.94** |
| SynCSE-scratch (SimCSE based) | 74.61 | 83.76 | 77.89 | 85.09 | 82.28 | 82.71 | 78.88 | 80.75 |

Table 4: Performance comparison of RoBERTa-base trained on various datasets, using the STS benchmark for evaluation. The reported metric is Spearman's correlation. The "†" symbol is used to indicate results reported in DINO. For SimCSE, we adopted the MNLI+SNLI dataset used in (Gao et al., 2021). "‡": GenSE released an NLI synthetic dataset comprising over 60 million samples. For a fair comparison, we randomly sampled from it the same number of samples used in the SimCSE dataset.

outperformed all the unsupervised baselines by more than 2 absolute points. Even when compared with supervised settings, our approach achieved performance near that of manual annotation on RoBERTa-base, falling behind by only about 1 point on RoBERTa-large. It's worth noting that while the supervised SimCSE training dataset (SNLI) and STS test data share a significant overlap in domains (for instance, both STSb and SNLI extensively used Flicker30k data (Plummer et al., 2015)), the domains were not explicitly known while generating the SynCSE-scratch dataset. Interestingly, SynCSE-partial does not always beat SynCSE-scratch as demonstrated in the RoBERTa-large case, which implies the potential of SynCSE-scratch as a promising approach to learn sentence embeddings without using any real data samples. By augmenting annotated NLI

data with the SynCSE-scratch synthetic dataset, our approach outperformed sup-SimCSE significantly, reaching a performance of 84.37% with RoBERta-large, suggesting that our synthetic data is complementary to human-labeled NLI datasets. "PromptCSE+EH" (Jiang et al., 2022b) achieves competitive performance in the supervised setups. As an orthogonal contribution, however, SynCSE may be combined with the loss function they proposed to further advance the results.

### 3.4 Reranking

Table 3 shows the results of the reranking tasks. Compared to the STS task, the domain of the reranking task data is more divergent from that of the NLI data used for training, as a result, SynCSE-scratch actually outperforms SynCSE-partial significantly, which implies the advantage of SynCSE-

| Model | Method | STS12 | STS13 | STS14 | STS15 | STS16 | STSb | SICK-R | Avg |
|---|---|---|---|---|---|---|---|---|---|
| RoBERTa-base | ZeroGen | 51.68 | 71.45 | 58.80 | 67.04 | 70.04 | 65.00 | 66.88 | 64.41 |
| | SynCSE-scratch (SimCSE based) | 71.81 | 83.43 | 76.90 | 83.39 | **82.33** | **82.89** | 77.39 | 79.73 |
| RoBERTa-large | ZeroGen | 50.97 | 70.90 | 59.97 | 69.59 | 68.79 | 65.43 | 65.72 | 64.48 |
| | SynCSE-scratch (SimCSE based) | **74.61** | **83.76** | **77.89** | **85.09** | 82.28 | 82.71 | **78.88** | **80.75** |

Table 5: Performance comparison of SynCSE-scratch and ZeroGen, using the STS benchmark for evaluation. The Spearman's correlation is reported.

| Model | Method | BIOSSES (Spearman's correlation) | StackOverflowDupQuestions (Mean Average Precision) |
|---|---|---|---|
| RoBERTa-base | unsup-SimCSE (Wikipedia domain) | 68.86 | 39.25 |
| | SynCSE-scratch (SimCSE based) | **80.12** | 43.22 |
| RoBERTa-large | unsup-SimCSE (Wikipedia domain) | 71.96 | 42.56 |
| | SynCSE-scratch (SimCSE based) | 77.73 | **45.67** |

Table 6: Performance comparison of the RoBERTa trained on the Wikipedia domain (using the publicly available unsup-SimCSE checkpoint) and specialized domains data generated by SynCSE-scratch.

scratch when in-domain unlabeled sentences are unavailable. SynCSE-scratch also surpasses other unsupervised baselines while SynCSE-partial underperforms them. Moreover, the combination of SynCSE-scratch with manually annotated datasets still facilitates further performance enhancement, substantiating that our method can aid in augmenting existing datasets.

## 3.5 Comparison with Other Synthetic Datasets

In addition to comparing with the MNLI+SNLI datasets used in SimCSE, we also compare our method with three other baselines that leverage synthetic NLI data: (1) GENSE (Chen et al., 2022) aims to automatically annotate the positive and hard negative examples with a LLM trained on existing NLI labeled dataset. We sample the same number of examples from their published dataset as those used in SynCSE; (2) The objective of DINO (Schick and Schütze, 2021) is to generate synthetic data for sentence embeddings as well. In DINO's most effective configuration, they generate the positive or hard negative samples and assign a similarity score to them based on the prompts used. As they have not made an NLI-style dataset available, we directly report results from their paper, and (3) ZeroGen (Ye et al., 2022a) proposes an efficienty unsupervised dataset generation method. We selected those examples that have been provided in both "entailment" and "not_entailment" sentences to construct sentence pairs, totaling 46,311 pairs, as training data. To ensure a fair comparison, we randomly sampled an equal number of examples gen-

erated by SynCSE-scratch. We compare the generated sentences of our methods with them in Table 11. From the table, we can find that our generated sentence can generate more diverse annotations. As depicted in Table 4, both SynCSE-scratch and SynCSE-partial have achieved performance on the STS task that surpasses that of DINO, GenSE. In a practical setting when generating a dataset from scratch (SynCSE-scratch), we compare our method with ZeroGen (Table 5), and the results show our method significantly outperforms the baseline.

## 3.6 Applying to Specialized Domains

SynCSE is advantageous when dealing with specialized domains where unlabeled data is unavailable. In such cases, traditional methods are not directly applicable. To evaluate SynCSE in this scenario, we conduct experiments on two another datasets focused on specialized domains – the BIOSSES (Soğancıoğlu et al., 2017) dataset of a semantic textual similarity task for the biomedical domain, and the StackOverflowDupQuestions (Liu et al., 2018) dataset of a reranking task for the programming questions domain. Specifically, our experimental design is based on the assumption that we only have access to the names of the target domains (i.e., "biomedicine" and "Stack Overflow website") without any data available. We run SynCSE-scratch in these settings. Concretely, we first generate 37k unlabeled sentences in the respective domain following the procedure described in Section §2.4, then generate positive and hard negatives for these sentences, and train the models. We use the publicly available unsupervised

| Model | Method | MR | CR | SUBJ | MRQA | SST | TREC | MRPC | Avg |
|-------|--------|----|----|------|------|-----|------|------|-----|
| | *Unsupervised methods* | | | | | | | | |
| RoBERTa-base | unsup-SimCSE[†] | 83.37 | 87.76 | **95.05** | 87.16 | 89.02 | **90.80** | 75.13 | 86.90 |
| | L2P-CSR[♠] | 79.67 | 88.30 | 94.27 | 87.70 | 87.50 | 81.14 | 76.47 | 85.01 |
| | PCL[‡‡] | 81.83 | 87.55 | 92.92 | 87.21 | 87.26 | 85.20 | 76.46 | 85.49 |
| | PrompRoBERTa[††] | 83.82 | 88.72 | 93.19 | **90.36** | 88.08 | 90.60 | 76.75 | 87.36 |
| | ConPVP[•] | 82.44 | 88.30 | 93.20 | 88.74 | 87.70 | 87.33 | 76.15 | 86.27 |
| | SynCSE-partial (SimCSE based) | 85.41 | **91.44** | 93.39 | 89.91 | **91.21** | 84.40 | **76.87** | **87.52** |
| | SynCSE-scratch (SimCSE based) | **85.47** | **91.44** | 92.53 | 89.67 | 90.94 | 81.60 | 76.06 | 86.82 |
| RoBERTa-large | unsup-SimCSE[†] | 84.66 | 88.56 | **95.43** | 87.50 | 89.46 | **95.00** | 72.41 | 87.57 |
| | L2P-CSR[♠] | 80.12 | 88.53 | 94.07 | 88.92 | 87.04 | 83.05 | 76.84 | 85.51 |
| | PCL[‡‡] | 84.47 | 89.06 | 94.60 | 89.26 | 89.02 | 94.20 | 74.96 | 87.94 |
| | SynCSE-partial (SimCSE based) | 87.18 | 92.02 | 94.16 | 90.76 | 91.65 | 86.80 | **76.87** | **88.49** |
| | SynCSE-scratch (SimCSE based) | **87.24** | **92.16** | 93.75 | **90.81** | **91.87** | 84.00 | 76.29 | 88.02 |
| | *Supervised methods* | | | | | | | | |
| RoBERTa-base | sup-SimCSE[†] | 85.08 | **91.76** | 94.02 | 89.72 | 92.31 | **91.20** | 76.52 | 88.66 |
| | sup-SimCSE | 85.05 | 90.97 | 94.20 | 89.37 | 91.49 | 88.60 | 76.87 | 88.08 |
| | PrompRoBERTa[††] | **85.74** | 91.47 | **94.81** | 90.93 | **92.53** | 90.40 | **77.10** | **89.00** |
| | SynCSE-scratch + SimCSE_NLI | 85.51 | 91.52 | 93.33 | 89.87 | 92.48 | 83.40 | 76.06 | 87.40 |
| RoBERTa-large | sup-SimCSE[†] | 88.12 | 92.37 | 95.11 | 90.49 | 92.75 | **91.80** | 76.64 | 89.61 |
| | sup-SimCSE | 87.89 | **92.61** | **95.20** | 90.77 | 92.86 | 90.80 | **77.22** | **89.62** |
| | SynCSE-scratch + SimCSE_NLI | **88.22** | 92.56 | 94.76 | **90.98** | **93.08** | 88.00 | 76.81 | 89.20 |

Table 7: Transfer task results of different sentence embedding models (measured as accuracy). The labels "unsup-" and "sup-" correspond to unsupervised and supervised settings, respectively. "[†]": results from (Gao et al., 2021); "[♠]": results from (Zhou et al., 2023); "[‡‡]": results from (Wu et al., 2022a); "[††]": results from (Jiang et al., 2022a); "[•]": results from (Zeng et al., 2022). The term "SynCSE-scratch + SimCSE_NLI " represents our synthetic data combined human labeled NLI dataset used in SimCSE.

| Method | STS12 | STS13 | STS14 | STS15 | STS16 | STSb | SICK-R | Avg |
|--------|-------|-------|-------|-------|-------|------|--------|-----|
| Naive Generation | 64.65 | 75.86 | 62.94 | 72.79 | 71.61 | 72.76 | 71.57 | 70.31 |
| SynCSE-scratch | **70.89** | **83.79** | **76.48** | **83.28** | **81.97** | **82.36** | **76.14** | **79.27** |

Table 8: Performance comparison of our synthetic dataset generation and the "Naive Generation" method.

SimCSE model checkpoint that was trained on the Wikipedia domain for comparison. This is because we assumed no access to unlabeled sentences in these domains, which is a practical setting. Our observations (Table 6) show that SynCSE-scratch outperforms the unsupervised SimCSE baseline significantly in both domains. This experiment further demonstrates the superiority of our method on new domains where no data is available – traditional unsupervised approaches like SimCSE tend to experience a domain transfer drop in performance in such scenarios.

## 3.7 Analysis

In this subsection, we provide an in-depth analysis of SynCSE. All results presented here are based on the RoBERTa-base model.

**Transfer tasks:** Following SimCSE, we execute seven transfer learning tasks: MR (Pang and Lee, 2005), CR (Hu and Liu, 2004), SUBJ (Pang and Lee, 2004), MPQA (Wiebe et al., 2005), SST-2 (Socher et al., 2013), TREC (Voorhees and Tice, 2000), and MRPC (Voorhees and Tice, 2000). These experiments are carried out with the same settings as used in SimCSE. As shown in Table 7, SynCSE-partial outperforms all unsupervised baselines.

**Comparion with the naive generation process:** To validate the effectiveness of our data synthesis process, we conduct an ablation experiment, where (1) we do not specify topics or genres when generating unlabeled sentences, and (2) we do not vary the prompt and exemplars but fix them the same (that are randomly selected from the pools) when generating the positive and hard negative labels.

|  | Counselor 1 | H2 | H3 | H4 | H5 | Avg |
|---|---|---|---|---|---|---|
| Fraction of ethically unsafe data | 1% | 0% | 0% | 1% | 0% | 0.4% |

Table 9: The result of the fraction of ethically unsafe data annotated by one psychological counselor and four postgraduate students. H* means the index of postgraduate annotators.

Other settings are kept the same as in SynCSE-scratch. We perform the ablation experiment on 22k examples. We denote the baseline without diversity control as "Naive Generation" and show them in the Table 8, our method outperforms the Naive Generation baseline by an average of 8.96%, demonstrating the critical role of diversity control in our data synthesis process.

**Ethical considerations of the synthetic dataset:** To evaluate the safety of our synthetic dataset, we ask five annotators (one of which is a psychological counselor and the other four are postgraduate students) to annotate whether the generated sentences have ethical problems. Specifically, we randomly select 100 sentences from those generated by SynCSE-scratch, and each sentence is independently evaluated by the five people for potential ethical problems. As the Table 9 suggests, only a minor portion of the data is classified as ethically unsafe, indicating that our synthetic dataset upholds a certain level of safety concerning ethical issues. This is not surprising since ChatGPT, the backend in our experiments, is already heavily aligned to avoid producing text with ethical or safety issues.

## 4 Related Work

Prior approaches for sentence embedding fall into two main categories: (1) supervised learning with labeled sentences, and (2) unsupervised sentence embedding with unlabeled sentences. Among these, works based on contrastive learning have proven to be the most effective. For unsupervised methods, SimCSE uses dropout masks to construct positive pairs for learning, while negative examples use in-batch negative examples. Some works employ data augmentation techniques on input sentences, such as word repetition (Wu et al., 2022b), case flipping (Wang et al., 2022c), or a combination of multiple data augmentation strategies to offset the bias caused by mono-augmentation (Wu et al., 2022a). PromptBERT (Jiang et al., 2022a)

uses prompts instead of the $[CLS]$ token to extract embeddings.

However, these unsupervised methods significantly lag behind their supervised counterparts. Supervised approaches usually derive positive and hard negative samples from labeled NLI datasets (Wang and Lu, 2022; Gao et al., 2021; Jiang et al., 2022a), but these datasets are limited in quantity and domain. Additionally, annotating a new NLI dataset is costly, especially in fields that require trained annotators. Chen et al. (2022) trained a T5 (Chung et al., 2022) model capable of producing positive and hard negative samples, while Ye et al. (2022b) implemented a continuously updated model to modify prompts for generation. However, the performance of these algorithms is still constrained by the performance of generators, which need labeled NLI data for training. Differing from these methods, which necessitate training an additional model, Wang et al. (2022b) proposed a rule-based algorithm capable of generating hard negative annotations. However, its diversity is limited to the prescribed rules. Gilardi et al. (2023) used ChatGPT for dataset annotation. However, their exploration was limited to tasks with explicit answer labels such as "RELEVANT" or "IRRELE-VANT". They did not attempt to annotate datasets that required diverse responses. Schick and Schütze (2021) also propose to generate both annotations and unlabeled sentences, while they do not focus on the contrastive learning framework.

## 5 Discussion

In this work, we propose SynCSE, a novel contrastive learning framework for learning sentence embeddings with synthetic data. We prompt LLMs to synthesize unlabeled sentences and their positive and hard negative examples. Furthermore, by utilizing example and prompt pools, we can specify the genre and topic of generated sentences, thereby enhancing the quality of the synthetic dataset. Experiments on both sentence similarity and reranking tasks demonstrate the effectiveness of SynCSE. The performance of SynCSE in this study strongly suggests the potential of synthetic datasets generated by the increasingly advanced LLMs of today. We envision that, through the effective use of prompting strategies with LLMs, synthetic datasets produced by these models could potentially serve as promising alternatives to real-world data across a wide range of tasks.

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

## A Prompt pools

In order to increase the diversity of input prompts, we designed a variety of prompts for generating positive samples, hard negative samples, and unlabeled data, which are adopted during generation based on certain probabilities. The specific prompts are displayed in Tables 12, 1, and 15. Given that generating image captions differs somewhat from generating other types of text, we have designed unique prompts for image captions to further enhance diversity, as illustrated in Table 16.

## B Genres and Topics

**Genres:** When generating unlabeled sentences, to make the newly generated sentences as different as possible from existing data, we specify the genre and topic of the new sentences. As long as the genre and topic of the new sentences are different from existing ones, the probability of these new sentences providing more new information to the dataset becomes higher. In this paper, we use 20 different genres (Table 10) and 31 different topics (Table 2). Before generating sentences, we use GPT-4 to generate 30 examples for each genre as one-shot example sentences. When using them, we first specify a genre and fill it into the "$[the\ description\ of\ the\ genre]$" in the prompt of Table 15, then randomly choose 6 from the topic list to fill into "$[topic_i]$". These descriptions are adapted from the genre specifications provided by GPT-4, thus, creating new descriptions does not require a significant effort.

**Topics:** We leveraged GPT-4 to generate an array of diverse topics, and 37 of these were randomly selected as the thematic grounding for our generation of unlabeled sentences. Concretely, these themes are: nature, technology, food, sports, culture, history, animals, environment, politics, finance, education, social issues, global issues, entertainment, healthcare, war, mathematical and electrical engineering, crime, relationships and emotional bonds, magic and mythical creatures, personal life stories, business strategies, fitness and mental health, global warming and conservation, various forms of art and cultural practices, teaching methodologies and learning styles, recipes and culinary techniques, ethical dilemmas and existential questions, space exploration and celestial phenomena, legal issues and courtroom drama, examination of past events and civilizations, ancient myths and legends,

scientific theories, life stories of notable individuals, COVID-19, immigration policies, and mental health.

| | **Genre descriptions** |
|---|---|
| 1 | in-person conversations |
| 2 | letters |
| 3 | reports, speeches, letters, and press releases from public domain government websites |
| 4 | fiction |
| 5 | image descriptions |
| 6 | video descriptions |
| 7 | news from websites or newspapers |
| 8 | reviews and critiques for shopping |
| 9 | headlines of news |
| 10 | dialogue of technical or Instructional tutorial |
| 11 | informative and expository texts which provide guidance or explanations related to a wide range of topics |
| 12 | STEM examination questions |
| 13 | travelogue |
| 14 | historical description |
| 15 | plots involving political intrigue and maneuvering |
| 16 | formal writings that present research findings or scholarly discussion |
| 17 | speeches given by politicians, often with the intent of persuading or informing an audience about political topics |
| 18 | written works such as poetry, drama, and novels |
| 19 | pieces of content shared on social media platforms |
| 20 | short image captions |
| 21 | messages paid for by a business or individual to promote a product, service, or event |

Table 10: The list of genre descriptions.

## C Hyperparameters

We employed gpt-3.5-turbo-0301 for sentence generation. For the generation of unlabeled sentences, we set the temperature to 1.3, top_p to 1.0, and both presence_penalty and frequency_penalty to 0.3. The input prompts were one-shot prompts; in the example, 10 sentences were generated at once, and during the generation process, 20 sentences were generated at once. During the generation of positive sample annotations, we set the temperature to 1.0 and top_p to 0.9. In the generation of neg-

| Method | Entailment | Contradiction |
|---|---|---|
| | ***Input:*** *A young man is getting ready to release a red kite.* | |
| DINO | A young man releasing a red kite.
A man getting ready to release a red kite. | A red kite releasing a red kite.
It was a big deal to him and he didn't know how he would explain it to his parents |
| GenSE | The man is prepared to fly the kite.
A man is planning to fly a kite. | A man is playing basketball.
The woman is flying a kite |
| SynCSE-scratch(ours) | A young man is preparing to let go of a red kite.
A man prepares to fly a crimson kite. | A young man getting ready to release a blue kite.
A young man gets ready to catch a red kite that has been released. |
| | ***Input:*** *One of the hotel's rooms* | |
| DINO | The hotel room
One of the hotel rooms | The other one is on fire
I have no idea what that is. |
| GenSE | A room inside a hotel.
A hotel room. | It's not the hotel's room.
There is no room at the hotel. |
| SynCSE-scratch(ours) | A room in the hotel.
A hotel room. | None of the hotel's rooms.
All of the hotel's rooms were fully booked for the weekend. |

Table 11: Comparison of different data synthesis methods. For samples of DINO and GenSE, we cite the generation sentences reported in (Ye et al., 2022b).

| **Positive prompts pools** |
|---|
| **Prompt1**: Please paraphrase the input sentence or phrase, providing an alternative expression with the same meaning. |
| **Prompt2**: Rewrite the following sentence or phrase using different words and sentence structure while preserving its original meaning. |
| **Prompt3**: Create a sentence or phrase that is also true, assuming the provided input sentence or phrase is true. |
| **Prompt4**: Please provide a concise paraphrase of the input sentence or phrase, maintaining the core meaning while altering the words and sentence structure. Feel free to omit some of the non-essential details like adjectives or adverbs. |

Table 12: Positive prompts pools. During the generation of positive samples, a prompt is sampled with a certain probability and inserted into the few-shot input prompts in Table 2, which are input in the form of multi-turn dialogues.

| | Unlabeled Sentence | Positive Label | Hard Negative Label | All |
|---|---|---|---|---|
| cost
(% per sentence) | 0.00007 | 0.00067 | 0.00076 | 0.0015 |

Table 13: The cost analysis of our method generating sentences with gpt-3.5-turbo.

ative sample annotations, we set the temperature to 1.0 and top_p to 0.95. Both positive and negative sample generations were 5-shot. Our training framework is based on SimCSE, which forcibly truncates parts of the sentence exceeding 32 words during training. To maintain a fair comparison, we filter out sentences with more than 32 words before training with the SimCSE framework after generating sentences with SynCSE-scratch.

# D Cost of the synthesize data

We used gpt-3.5-turbo to synthesize data that is not very expensive, currently costing 0.0015 dollars per 1K tokens for input and 0.002 dollars per 1K tokens for output. Concretely, there are three parts in the data generation process: unlabeled sentences, positive labels, and hard negative labels. Since the length of each input varies, to quantify the cost, we randomly sampled 40 inputs and calculated the average cost per sentence. As detailed in Table 13, our method cost a total of around 1.5 $ for generating 1000 sentences, and the total cost of producing the 276k sentences used in our experiments of SynCSE-scratch in Table 2 is around 414 $. In the domain specialized task (Table 6), we just generate 37k sentence pairs and significantly surpass SimCSE in the target domain, and the cost is around 55 $. We would like to highlight that

| Data | 0% | 20% | 40% | 60% | 80% | 100% |
|---|---|---|---|---|---|---|
| **Avg. STS** | 82.04 | 82.10 | 82.73 | 82.58 | **82.75** | 82.61 |

Table 14: Performance of SimCSE_NLI when combined with varying amounts of our synthetic SynCSE-scratch dataset. We report the performance on the avg STS results on the test set.

the rate per sentence above is much cheaper than manually labeling data; for instance, in machine translation tasks, human translation (around $0.1 per word) can be thousands of times costlier than using gpt-3.5-turbo (Neubig and He, 2023).

# E    Synthetic data amount

We also analyzed the impact on performance when augmenting the volume of generated data on the manually curated dataset, as shown in Table 14. Since the domain of SynCSE-scratch is established upon its completion, the performance ceases to increase after a certain amount of SynCSE-scratch data is added to SimCSE. This may be due to the fact that the added data is randomly sampled, which likely already covers the domain of SynCSE-scratch.

| **Prompt pool for generating unlabeled sentences with specified genres and topics.** |
|---|
| **Prompt1**: Devise $[number]$ distinct and diverse sentences that may appear in $[the\ description\ of\ the\ genre]$, covering a range of subjects ($[topic_1]$, $[topic_2]$, $[topic_3]$, $[topic_4]$, $[topic_5]$, $[topic_6]$,etc.). These sentences should present a mix of complexity levels, from elementary structures akin to "Birds fly in the sky." to more sophisticated ones. Aim for a low degree of lexical overlap and an extensive vocabulary. Incorporate a variety of sentence modes - declarative, interrogative, exclamatory, imperative, and descriptive. The sentences should oscillate in tone between informative, persuasive, descriptive, and narrative, and should present varied perspectives. Vary the length of the sentences, ranging from concise phrases of 3-5 words to longer sentences containing 20-40 words. |
| **Prompt2**: Construct $[number]$ varied and diverse sentences that may appear in $[the\ description\ of\ the\ genre]$ encompassing a multitude of topics such as $[topic_1]$, $[topic_2]$, $[topic_3]$, $[topic_4]$, $[topic_5]$, $[topic_6]$ and so on. Aim for low lexical repetition and a rich vocabulary variety. Ensure to blend different sentence structures - declarative, interrogative, exclamatory, imperative, and descriptive. The sentences should convey different tones, including informative, persuasive, descriptive, and narrative, and should be expressed from various perspectives. Vary the length of the sentences, ranging from concise phrases of 3-5 words to longer sentences containing 20-30 words. |
| **Prompt3**: Create $[number]$ varied and diverse sentences that may appear in $[the\ description\ of\ the\ genre]$, spanning an array of topics such as $[topic_1]$, $[topic_2]$, $[topic_3]$, $[topic_4]$, $[topic_5]$, $[topic_6]$ and so on. Include a mix of sentence styles - declarative, interrogative, exclamatory, imperative, and descriptive. Vary the length of the sentences, ranging from concise phrases of 3-5 words to 25-35 words. |
| **Prompt4**: Compose $[number]$ assorted and diverse sentences that may appear in $[the\ description\ of\ the\ genre]$, touching upon various themes like $[topic_1]$, $[topic_2]$, $[topic_3]$, $[topic_4]$, $[topic_5]$, $[topic_6]$ and so on. The sentences should reflect a spectrum of complexity levels, from basic structures such as "The sun sets in the west." to more elaborate forms. Aspire for minimal lexical redundancy and a wide array of vocabulary. Incorporate a blend of sentence types - declarative, interrogative, exclamatory, imperative, and descriptive. The sentences should shift in tone, cycling between informative, persuasive, descriptive, and narrative, and should vary in perspective. Vary the length of the sentences, ranging from concise phrases of 3-5 words to longer sentences containing 25-45 words. |

Table 15: Prompt pool for generating unlabeled sentences with specified genres and topics. When generating unlabeled sentences, we randomly sample a prompt. Here, "$[number]$" indicates the number of sentences generated each time, "$[the\ description\ of\ the\ genre]$" provides a specific description of the given genre, and "$[topic_i]$" is a topic sampled from the topic list.

| **Prompt pool for generating image captions.** |
| --- |
| **Prompt1**: Please randomly generate $[number]$ diverse sentences in the style of $[the\ description\ of\ the\ genre]$, similar to image captions in the Flickr30k dataset, covering a wide range of subjects, actions, contexts, and settings. The sentences do not need to be semantically related. Please make sure to generate sentences with a uniform distribution in length, ranging from short phrases to longer ones. |
| **Prompt2**: Kindly generate $[number]]$ unique sentences reflecting the style of $[the\ description\ of\ the\ genre]$, drawing inspiration from the diverse scenarios found in the Flickr30k dataset. The sentences should cover an extensive array of themes, actions, surroundings, and situations. Please create a mix of simple sentences like "A man strums a guitar", along with more elaborate ones. |
| **Prompt3**: Kindly generate $[number]$ distinct sentences with simple structures, each may appear in $[the\ description\ of\ the\ genre]$. These sentences should touch on a wide array of topics, actions, and environments without necessarily having a semantic link. Strive to provide sentences of various lengths, from short to moderately long, all while maintaining simplicity and clarity. |
| **Prompt4**: Please randomly generate $[number]$ diverse and descriptive sentences which may appear in $[the\ description\ of\ the\ genre]$, ensuring they mirror the structure of the following image captions: $[example_1]$, $[example_2]$, $[example_3]$, $[example_4]$. Each sentence should vividly portray the primary activity in a hypothetical image or segment of a video, articulating the main subject(s), their actions, and any important objects or secondary subjects involved. |

Table 16: The Prompt pool for image caption generation. We designed a separate prompt for generating image captions to enhance diversity. Here, "$[number]$" denotes the number of sentences to generate, "$[the\ description\ of\ the\ genre]$" provides a description for generating captions, and "$[example_4]$" is a randomly sampled example sentence.