# OpenReview forum: "Contrastive Learning of Sentence Embeddings from Scratch"
_EMNLP/2023/Conference — EMNLP 2023 Main_

### Official Review · Reviewer_DJxQ · 2023-08-05

**Typos Grammar Style And Presentation Improvements:** Information (like year, conference na…
**Soundness:** 4

**Excitement:**

4: Strong: This paper deepens the understanding of some phenomenon or lowers the barriers to an existing research direction.

**Paper Topic And Main Contributions:**

This paper trains sentence embeddings in a contrastive learning manner, in which the contrastive training data is generated by LLM with many prompts in two mode. A series of experiments and analysis are conducted to prove the effectiveness of the work.

**Questions For The Authors:**

How to ensure the quality of synthetic data? What is the volume?
What is the improvement compared with Roberta or other pre-trained encoders like BERT without additional sentence embedding learning?
What is the generalization of this embedding method to other languages?

**Reasons To Accept:**

The use of LLM in specific scenarios is a new trend in NLP and the authors designed lots of prompts.
There is an amount of experiments and have better performance than other models and other synthetic data.
Writing of the paper is satisfying.

**Reasons To Reject:**

Novelty of the work is limited. It seems that the core lies in the usage of LLM to generate training data.

**Reproducibility:**

4: Could mostly reproduce the results, but there may be some variation because of sample variance or minor variations in their interpretation of the protocol or method.

**Reviewer Confidence:**

4: Quite sure. I tried to check the important points carefully. It's unlikely, though conceivable, that I missed something that should affect my ratings.

---

> ### Author Rebuttal · Authors · 2023-08-29
>
> We appreciate your insightful feedback. Here are our responses to your concerns:
>
> ### Q1:The novelty of our paper.
>
> We would like to highlight two main novel contributions of this work:
>
> 1. Using LLMs to generate data is far from enough and our carefully designed synthesis methods play an important role – we designed a top-down synthesis scheme to ensure diversity, where we first select topics/genres (Section 2.4, Appendix B), sample prompts and exemplars (Section 2.3, Appendix A), and then produce the final data examples. Such a diversity control is in fact a critical design of SynCSE. To validate the effectiveness of our diversity control, we conduct an ablation experiment, where (a)  we do not specify topics or genres when generating unlabeled sentences, and (b) we do not vary the prompt and exemplars but fix them the same when generating the positive and hard negative labels.  Other settings are kept the same as in SynCSE-scratch. We perform the ablation experiment on 22k examples based on Roberta-base, and the results are shown below. We denote the baseline without diversity control as “Naive Generation”.
>
> | Method       | STS12 | STS13 | STS14 | STS15 | STS16 | STSb | SICK-R | Avg |
> |:-------------|:----------------|:---------|:---------|:---------|:---------|:---------|:--------|:----------|
> | Naive Generation |    64.65 | 75.86 | 62.94 | 72.79 | 71.61 |    72.76     |    71.57   | 70.31          |
> |     SynCSE-scratch   |    70.89  | 83.79 |  76.48  |  83.28  |  81.97  |  82.36   |  76.14  |  79.27      |
>
> Our method outperforms the Naive Generation baseline by an average of 8.96 points, demonstrating the critical role of diversity control in our data synthesis process, which is one of the key novelties presented in this work.
>
> 2. Combining ChatGPT with specially designed prompting strategies (e.g. the diversity control as described above), we were able to push the performance of synthetic datasets to a new level, greatly outperforming traditional unsupervised methods and approaching the performance of supervised methods. Our results first mark that data synthesis from LLMs could be **practically** useful and superior to using real unlabeled data. We think that this is a non-trivial and “novel” empirical contribution compared to previous work.
>
>
>
> ### Q2: How to ensure the quality of the data.
>
>
> For diversity, we ensure the diversity of the synthetic dataset primarily through two strategies: 1) Manually drafting and specifying the topics and genres (Appendix B) when generating data, so that the newly generated data's topics and genres differ from existing data to enhance diversity. 2) Constructing prompt and example pools, and varying the prompts and exemplars during generation to increase diversity;
> For quality,  we found that ChatGPT’s performance in generating unlabeled sentences and positive/negative labels is reasonably good and satisfying by manually investigating a few examples. We just filter out unlabeled sentences of more than 32 words to match the max sequence length used in SimCSE (line 931)
>
>
>
>
> ### Q3: The volume of the generated dataset.
> For a fair comparison, we generated a dataset similar in size to the one used in the supervised experiments with SimCSE (around 276K sentence pairs).
>
> ### Q4:  The improvement compared with BERT without sentence embeddings learning.
>
> SimCSE (Table 5) provides the performances in comparison with a BERT model that does not include sentence embeddings learning.
>
> ### Q5:  The generalization of this embedding method to other languages
> That is a good point. ChatGPT supports multiple languages and can directly output in various languages; however, we have not conducted any experiments in this regard. We can explore this aspect as part of our future work.
>
> ### Q6: Typos
> Thanks for catching those typos! We will fixe them in the next revision of the paper.
>
> We will add these experiments to the next revision of the paper.

---

### Official Review · Reviewer_YEGU · 2023-08-05

**Soundness:** 4

**Excitement:**

4: Strong: This paper deepens the understanding of some phenomenon or lowers the barriers to an existing research direction.

**Missing References:**

NA

**Paper Topic And Main Contributions:**

This paper discusses a novel approach for improving unsupervised sentence representation learning using contrastive learning methods. Sentence embeddings are vector representations of sentences that capture their semantic meaning, and contrastive learning is a popular technique used to learn meaningful representations from data. The authors propose a new contrastive learning framework called SynCSE. This framework leverages large language models to synthesize the required data samples for contrastive learning. The experimental results demonstrate the effectivenss of the proposed framework.

**Questions For The Authors:**

a) Can you provide more details on the ethical considerations taken into account during the synthesis of data using large language models? How do you ensure that the generated synthetic data is free from biases and representative of the underlying distribution?

b) How does the computational cost of generating synthetic data using large language models impact the feasibility and scalability of the proposed approach? Have you conducted any analyses to quantify the resources required for training the models and generating the synthetic data?

**Reasons To Accept:**

a) The paper addresses the issue of data availability, particularly in domains where labeled or large-scale unlabeled data is scarce or hard to obtain. By proposing a method to create synthetic data samples, it provides a practical solution for training effective sentence embeddings.

b) Since the SynCSE approach is based on synthetic data generation, it is not restricted to specific domains or data types. This makes it potentially applicable to a wide range of NLP tasks, even in domains where acquiring large-scale unlabeled data is challenging.

**Reasons To Reject:**

a) The novelty of the proposed method is limited. There is an existing work [1] that has a very similar objective and methodology with this paper. The main difference between this paper and [1] is that this paper adopts ChatGPT, a much more powerful language model, to produce the synthetic data. More insights on how ChatGPT makes improvements over previous language models are needed.

b) Generating synthetic data using large language models can be computationally expensive and resource-intensive. The paper should provide insights into the computational requirements of the proposed approach, considering the scale of the language models used and the amount of data generated.

Reference
[1] Timo Schick and Hinrich Schütze. 2021. Generating datasets with pretrained language models. In Proceedings of EMNLP.

**Reproducibility:**

4: Could mostly reproduce the results, but there may be some variation because of sample variance or minor variations in their interpretation of the protocol or method.

**Reviewer Confidence:**

4: Quite sure. I tried to check the important points carefully. It's unlikely, though conceivable, that I missed something that should affect my ratings.

**Typos Grammar Style And Presentation Improvements:**

I think overall the paper is well-written and effectively communicates the research objectives, methodologies, and results.

---

> ### Author Rebuttal · Authors · 2023-08-29
>
> Thanks for your helpful comments on our work! Below we address your concerns:
>
> ### Q1: Novelty of this work compared to  DINO[1]
>
> We would like to highlight two main differences between this work and DINO:
>
> 1. The synthesis methods are essentially different – we designed a top-down synthesis scheme to ensure diversity, where we first select topics/genres (Section 2.4, Appendix B), sample prompts, and exemplars (Section 2.3, Appendix A), and then produce the final data examples. Such a diversity control is distinct from DINO, and in fact a critical design of SynCSE. To validate the effectiveness of our diversity control, we conduct an ablation experiment, where (a)  we do not specify topics or genres when generating unlabeled sentences, and (b) we do not vary the prompt and exemplars but fix them the same when generating the positive and hard negative labels.  Other settings are kept the same as in SynCSE-scratch. We perform the ablation experiment on 22k examples based on Roberta-base, and the results are shown below. We denote the baseline without diversity control as “Naive Generation”.
>
> | Method       | STS12 | STS13 | STS14 | STS15 | STS16 | STSb | SICK-R | Avg |
> |:-------------|:----------------|:---------|:---------|:---------|:---------|:---------|:--------|:----------|
> | Naive Generation |    64.65 | 75.86 | 62.94 | 72.79 | 71.61 |    72.76     |    71.57   | 70.31          |
> |     SynCSE-scratch   |    70.89  | 83.79 |  76.48  |  83.28  |  81.97  |  82.36   |  76.14  |  79.27      |
>
>
> Our method outperforms the Naive Generation baseline by an average of 8.96 points, demonstrating the critical role of diversity control in our data synthesis process, which is one of the key novelties compared to DINO.
>
> 2. As the reviewer pointed out, while DINO uses GPT-2 as the backend, we leverage a much stronger model, ChatGPT. Combining ChatGPT with specially designed prompting strategies (e.g. the diversity control as described above), we were able to push the performance of synthetic datasets to a new level, greatly outperforming traditional unsupervised methods and approaching the performance of supervised methods. Our results first mark that data synthesis from LLMs could be **practically** useful and superior to using real unlabeled data. We think that this is a non-trivial and “novel” empirical contribution compared to DINO.
>
>
>
>
> ### Q2: The cost of generating synthetic dataset and training the models
> We used gpt-3.5-turbo to synthesize data that is not very expensive, currently costing 0.0015 dollars per 1K tokens for input and 0.002 dollars per 1K tokens for output. Concretely, there are three parts in the data generation process: unlabeled sentences, positive labels, and hard negative labels. Since the length of each input varies, to quantify the cost, we randomly sampled 40 inputs and calculated the average cost per sentence:
> |                      | Unlabeled Sentence | Positive Label | Hard Negative Label | All |
> |:----------------------|:--------------------|:----------------|:---------------------|:-----|
> | cost($ per sentence) |        0.00007            |   0.00067            |         0.00076            |    0.0015 |
>
> This makes a total of 1.5 dollars for generating 1000 sentences, and the total cost of producing the 276k sentences used in our experiments of SynCSE-scratch in Table 1 is 414 dollars. In our response to Q2 of Reviewer P81C, we just generate 37k sentence pairs and significantly surpass SimCSE in the target domain, and the cost is 55 dollars.
> We would like to highlight that the rate per sentence above is much cheaper than manually labeling data; for instance, in machine translation tasks, human translation (around $0.1 per word) can be thousands of times costlier than using gpt-3.5-turbo [2].
>
> ### Q3: Ethical considerations of the synthetic dataset
>
> This is a good point. To evaluate the safety of our synthetic dataset. We ask five annotators (one of which is a psychological counselor and the other four are postgraduate students) to annotate whether the generated sentences have ethical problems. Specifically, we randomly select 100 sentences from those generated by SynCSE-scratch, and each sentence was independently evaluated by the five people for potential ethical problems. The results are as follows:
> |        | Human 1 (counselor)|Human 2| Human 3| Human 4|Human 5 | Avg |
> |:----------------------|:--------------|:--------------|:-------|:-------|:-------|:-------|
> | Fraction of ethically unsafe data |        1%        |           0%     |      0%       |   1%   | 0%  | 0.4%    |
>
> As the table suggests, only a minor portion of the data was classified as ethically unsafe, indicating that our synthetic dataset upholds a certain level of safety concerning ethical issues. This is not surprising since ChatGPT, the backend in our experiments, is already heavily aligned to avoid producing text with ethical or safety issues.
>
> We thank the reviewer for the advice and raising these good points, and we will add these additional results and analysis above to the next revision of the paper.
>
> Reference
>
>  [1] Schick, T., & Schütze, H. (2021, November). Generating Datasets with Pretrained Language Models. In Proceedings of the 2021 Conference on Empirical Methods in Natural Language Processing (pp. 6943-6951).
>  [2] Neubig, G., & He, Z. (2023). Zeno GPT Machine Translation Report

---

### Official Review · Reviewer_P81C · 2023-08-05

**Soundness:** 4

**Excitement:**

4: Strong: This paper deepens the understanding of some phenomenon or lowers the barriers to an existing research direction.

**Paper Topic And Main Contributions:**

The paper presents SynCSE, a contrastive learning framework that trains sentence embeddings using synthetic data. The authors explore two variants of SynCSE: SynCSE-partial, which synthesizes positive and hard negative examples given unlabeled sentences, and SynCSE-scratch, which generates sentences and their annotations from scratch. Experimental results show that both SynCSE-partial and SynCSE-scratch outperform unsupervised baselines and achieve comparable performance to supervised models in most settings.

**Questions For The Authors:**

1. Can you provide more details about the synthetic data generation process? How did you ensure the diversity and quality of the generated data?
2. Have you considered evaluating the performance of SynCSE on tasks that require specific domain knowledge? How do you think SynCSE would perform in such scenarios?

**Reasons To Accept:**

1. Innovative methodology: The use of large language models (LLMs) to synthesize data for contrastive learning is a novel approach.
2. Practical application: The framework addresses the challenge of acquiring labeled data in certain domains and provides a solution using synthetic data.
3. Comprehensive evaluation: The paper includes experiments on sentence similarity and reranking tasks, demonstrating the effectiveness of SynCSE in different settings.
4. Clear exposition: The paper provides a well-structured explanation of the proposed framework, making it easy to understand.

**Reasons To Reject:**

1. Limited comparison with other synthetic datasets: The comparison with GenSE and DINO is limited, and a more detailed analysis and comparison with other approaches would strengthen the paper.
2. Lack of evaluation on specialized domains: The paper does not evaluate the performance of SynCSE on tasks requiring specialized domain knowledge, which limits its applicability in certain fields.

**Reproducibility:**

3: Could reproduce the results with some difficulty. The settings of parameters are underspecified or subjectively determined; the training/evaluation data are not widely available.

**Reviewer Confidence:**

3: Pretty sure, but there's a chance I missed something. Although I have a good feel for this area in general, I did not carefully check the paper's details, e.g., the math, experimental design, or novelty.

---

> ### Author Rebuttal · Authors · 2023-08-29
>
> Thank you for your time and helpful comments! We address your concerns below:
>
>
> ### Q1: Limited comparison with other synthetic datasets
> Thank you for your advice! Methods based on contrastive learning have dominated this task, and they require NLI-style data. Therefore, we only compare with synthetic datasets that have NLI-style formats. During the rebuttal, we further compare our approach with the effective work of ZeroGen [1]. Specifically, we selected those examples that have been provided in both "entailment" and "not_entailment" sentences to construct sentence pairs, totaling 46,311 pairs, as training data. To ensure a fair comparison, we randomly sampled an equal number of examples generated by SynCSE-scratch. The experimental results are as follows:
>
> |                             | STS12 | STS13 | STS14 | STS15 | STS16 | STSb | SICK-R | Avg |
> |:-----------------------------|:-------|:-------|:-------|:-------|:-------|:------|:--------|:-----|
> | ZeroGen_RoBERTa_base             |      51.68 | 71.45 | 58.80 | 67.04 | 70.04 |    65.00     |      66.88      | 64.41      |
> | SynCSE-scratch_RoBERTa_base |  71.81 | 83.43 | 76.90 | 83.39 | 82.33 | 82.89 |  77.39 | 79.73      |
> | ZeroGen_RoBERTa_large             |    50.97 | 70.90 | 59.97 | 69.59 | 68.79 |    65.43     |      65.72      | 64.48     |
> | SynCSE-scratch_RoBERTa_large |     70.33 | 81.53 | 75.31 | 83.02 | 81.57 |    81.15    |      79.29      | 78.89    |
>
> Results show that our method consistently outperforms ZeroGen. This could be because ZeroGen is not designed specifically for this task, and there are significant differences in how to enhance data diversity. Additionally, in ZeroGen's implementation of generating QNLI, unlabeled sentences are sampled from a specific domain's corpus, which can also be costly. Conversely, our method only requires the target domain's name.
>
>
>
> ### Q2: Lack of evaluation on specialized domains
> Thanks for the advice! We conduct experiments on two another datasets focused on specialized domains – the BIOSSES[2] dataset of a semantic textual similarity task for the biomedical domain, and the StackOverflowDupQuestions[3] dataset of a reranking task for programming questions domain. Specifically, our experimental design is based on the assumption that we only have access to the names of the target domains (i.e., 'biomedicine' and 'Stack Overflow website') without any data available. We run SynCSE-scratch in these settings. Concretely, we first generate 37k unlabeled sentences in the respective domain following the procedure described in Section 2.4, then generate positive and hard negatives for these sentences, and train the models.
>
> The results are listed as follows:
>
> | Model     | Method     | BIOSSES (Spearman's correlation) |  StackOverflowDupQuestions  (Mean Average Precision) |
> |:-------------|:--------------|:------------------------------------|:----------------------------|
> |                             | unsup-SimCSE                  |     68.86        |       39.25    |
> | RoBERTa-base  | SynCSE-scratch      |   80.12  |  43.22    |
> |               | unsup-SimCSE                               |      71.96       |        42.56   |
> | RoBERTa-large | SynCSE-scratch        |  77.73     |     45.67  |
>
> We use the publicly available unsupervised SimCSE model checkpoint that was trained on the Wikipedia domain for comparison. This is because we assumed no access to unlabeled sentences in these domains, which is a practical setting. Our observations show that SynCSE-scratch outperforms the unsupervised SimCSE baseline significantly in both domains. This experiment further demonstrates the superiority of our method on new domains where no data is available – traditional unsupervised approaches like SimCSE tend to experience a domain transfer drop in performance in such scenarios. We will add these experiments to the next revision of the paper.
>
> ### Q3:  More details about the synthetic data generation process.
> Using SynCSE-scratch as an example, the complete data generation process includes two parts:
> 1. Performing one-shot generation to generate unlabeled sentences, specifying topics and genres as described in Appendix B and Table 8;
> 2. Generating positive/hard negative labels in a 5-shot manner, as illustrated in Figures 3 and 4.
> We keep the sentences less than 32 words following the hyperparameter of max sequence length in SimCSE during training (as described in line 931). The hyperparameters for generation are detailed in Appendix C. The version of the ChatGPT is: gpt-3.5-turbo-0301.
>
> ### Q4: How to ensure the diversity and quality of the generated data
>
> Indeed, this is one of our core contributions. Specifically,
>
> 1. For diversity, we ensure the diversity of the synthetic dataset primarily through two strategies: 1) Manually drafting and specifying the topics and genres (Appendix B) when generating data, so that the newly generated data's topics and genres differ from existing data to enhance diversity. 2) Constructing prompt and example pools, and varying the prompts and exemplars during generation to increase diversity;
>
> 2. For quality,  we found that ChatGPT’s performance in generating unlabeled sentences and positive/negative labels is reasonably good and satisfying by manually investigating a few examples. We just filter out unlabeled sentences of more than 32 words to match the max sequence length used in SimCSE (line 931)
>
> To validate the effectiveness of our data synthesis process, we conduct an ablation experiment, where (1)  we do not specify topics or genres when generating unlabeled sentences, and (2) we do not vary the prompt and exemplars but fix them the same (that are randomly selected from the pools) when generating the positive and hard negative labels.  Other settings are kept the same as in SynCSE-scratch. We perform the ablation experiment on 22k examples based on Roberta-base, and the results are shown below. We denote the baseline without diversity control as “Naive Generation”.
>
> | Method       | STS12 | STS13 | STS14 | STS15 | STS16 | STSb | SICK-R | Avg |
> |:-------------|:----------------|:---------|:---------|:---------|:---------|:---------|:--------|:----------|
> | Naive Generation |    64.65 | 75.86 | 62.94 | 72.79 | 71.61 |    72.76     |    71.57   | 70.31          |
> |     SynCSE-scratch   |    70.89  | 83.79 |  76.48  |  83.28  |  81.97  |  82.36   |  76.14  |  79.27      |
>
> Our method outperforms the Naive Generation baseline by an average of 8.96 points, demonstrating the critical role of diversity control in our data synthesis process.
>
> Reference
>
> [1] Ye, J., Gao, J., Li, Q., Xu, H., Feng, J., Wu, Z., ... & Kong, L. (2022, December). ZeroGen: Efficient Zero-shot Learning via Dataset Generation. In Proceedings of the 2022 Conference on Empirical Methods in Natural Language Processing (pp. 11653-11669).
> [2] Soğancıoğlu, G., Öztürk, H., & Özgür, A. (2017). BIOSSES: a semantic sentence similarity estimation system for the biomedical domain. Bioinformatics, 33(14), i49-i58.
> [3] Liu, X., Wang, C., Leng, Y., & Zhai, C. (2018, November). Linkso: a dataset for learning to retrieve similar question answer pairs on software development forums. In Proceedings of the 4th ACM SIGSOFT International Workshop on NLP for Software Engineering (pp. 2-5).

---

### Meta-Review · Area_Chair_jgEW · 2023-09-19

**Recommendation:** 4

**Metareview:**

The paper introduces SynCSE, a contrastive learning framework for training sentence embeddings using synthetic data. It explores two variants: SynCSE-partial, which generates positive and hard negative examples from unlabeled sentences, and SynCSE-scratch, which creates sentences and their annotations from scratch. Experimental results show that both variants outperform unsupervised baselines and perform competitively with supervised models in most settings.

Reviewers have all found that the paper is sound and strong, therefore, I recommend acceptance of this paper.

---

### Decision · Program_Chairs · 2023-10-07

**Decision:**

Accept-Main

**Comment:**

The paper introduces SynCSE, a contrastive learning framework for training sentence embeddings using synthetic data. It explores two variants: SynCSE-partial, which generates positive and hard negative examples from unlabeled sentences, and SynCSE-scratch, which creates sentences and their annotations from scratch. Experimental results show that both variants outperform unsupervised baselines and perform competitively with supervised models in most settings.

Reviewers have all found that the paper is sound and strong, therefore, I recommend acceptance of this paper.